# Determining Strawberries' Varying Maturity Levels by Utilizing Image Segmentation Methods of Improved DeepLabV3+

Changqing Cai [1,2], Jianwen Tan [3] , Peisen Zhang [4], Yuxin Ye [5,*] and Jian Zhang [6,7,*]

1. Engineering Training Center, Changchun Institute of Technology, Changchun 130012, China
2. National and Local Joint Engineering Research Center for Smart Distribution Network Measurement, Control and Safe Operation Technology, Changchun 130012, China
3. College of Electrical and Information Engineering, Changchun Institute of Technology, Changchun 130012, China
4. College of Energy and Power Engineering, Changchun Institute of Technology, Changchun 130012, China
5. College of Computer Science and Technology, Jilin University, Changchun 130012, China
6. Faculty of Agronomy, Jilin Agricultural University, Changchun 130018, China
7. Department of Biology, University of British Columbia, Okanagan, Kelowna, BC V1V 1V7, Canada
* Correspondence: yeyx@jlu.edu.cn (Y.Y.); jian.zhang@ubc.ca (J.Z.)

**Abstract:** Aiming to determine the inaccurate image segmentation of strawberries with varying maturity levels due to fruit adhesion and stacking, this study proposed a strawberry image segmentation method based on the improved DeepLabV3+ model. The technique introduced the attention mechanism into the backbone network and the atrous spatial pyramid pooling module of the DeepLabV3+ network, adjusted the weights of feature channels in the neural network propagation process through the attention mechanism to enhance the feature information of strawberry images, reduced the interference of environmental factors, and improved the accuracy of strawberry image segmentation. The experimental results showed that the proposed method can accurately segment images of strawberries with different maturities; the mean pixel accuracy and mean intersection over union of the model were 90.9% and 83.05%, respectively, and the frames per second (FPS) was 7.67. The method can effectively reduce the influence of environmental factors on strawberry image segmentation and provide an effective approach for accurate operation of strawberry picking robots.

**Keywords:** improved DeepLabV3+; attention mechanism; image segmentation; strawberry

## 1. Introduction

Strawberry has been planted worldwide due to its strong environmental adaptability and high economic benefits, among which China ranks first in the world in terms of strawberry cultivation area [1]. With the increasing scale of strawberry cultivation, the traditional manual picking can no longer meet the harvesting/picking demand of the strawberry industry, and the automation of strawberry picking has become the focus of research in many countries [2]. Strawberry ripeness determination is used as a judgment condition for automated strawberry picking.

Researchers around the world have investigated various techniques to determine strawberry ripeness classes, including spectroscopic techniques and machine vision [3]. Spectroscopy mainly used optical information to obtain information about the chemical composition and physical properties of strawberry fruit, which is used as input data for classification models to determine strawberry fruit ripeness classes. Raj et al. [4] used a narrowband hyperspectral radiometer to collect the reflection characteristics of strawberry fruits of different maturity levels and subsequently took the drying method to obtain the water content of ground strawberries. They used the above two sets of data as input data for a linear support vector machine model. It was able to achieve 98% accuracy in strawberry

maturity classification; The classification accuracy of strawberry maturity is 71% when only the data set of strawberry fruit moisture content was available. Su [5] used a hyperspectral imager to collect one-dimensional spectral and three-dimensional spectral images and used the above data as input data in the residual network to build a one-dimensional model. Constructed networks for strawberry fruit classification at different maturity levels showed the accuracy of both networks reached 84%. However, this paper mainly adopted machine vision to obtain the image information of strawberry to determine the ripeness of strawberry fruit. Meanwhile, the strawberry fruit is in an unstructured environment, and there are factors such as dense distribution among fruits, leaf shading, and fruit stacking which make it difficult for the existing image recognition algorithms to perform accurately. Therefore, the research on strawberry image recognition, segmentation and localization methods with high recognition accuracy and strong environmental adaptability is the focus of this paper.

In recent years, as the research on deep learning technology in the field of computer vision has become very active [6,7], its recognition and segmentation of strawberry images has gradually become a hot topic for domestic and international research [8]. Semantic segmentation and instance segmentation are more popular in deep learning segmentation, and semantic segmentation networks mainly use models such as FCN [9,10], PSPNet [11], U-Net [12], SegNet [13], and DeepLab [14–16], while instance segmentation often uses Mask R-CNN [17–20] models. Ilyas [21] proposed a novel convolutional encoder/decoder network model, which combines dilated residual blocks (DRB), bottleneck blocks (BB) and an adaptive receptive field module (ARFM), reduces the network computational complexity, and enhances the network feature extraction capability. However, because the dataset is too small and usually does not contain occluded strawberry images, it is still necessary to increase the number of strawberry images and enrich the diversity of strawberry images to verify the general applicability of the model. Yu [22] proposed to use the Mask R-CNN model for recognition and segmentation of strawberry images in unstructured environments, combining Resnet50 with Feature Pyramid Network (FPN) as the backbone network to extract feature maps, then inputting the feature maps to a Region Proposal Network (RPN) to generate a Region of Interest (ROI), and finally inputting the region of interest to FCN; the target mask image was generated and the experimental results of this method for the masked strawberry image showed that the average detection accuracy is 95.78%; Jia [23] used the U-Net model as a prototype, and selected the convolution with the same improved VGG16 model to extract image feature information and retain more feature information to improve the model segmentation accuracy, and the experimental results of this method for obscured strawberry images showed that the average detection accuracy is 96.05%; Ge [24] proposed to use the Mask R-CNN model with DCNN network backbone to segment strawberry images. For the strawberry fruit occlusion problem, the occluded strawberries were detected and the occluded part was compensated by the enclosing box; the experimental results of this method on the occluded strawberry images showed that the average detection accuracy was 94%. The models in the literature [22–24] were focused on high detection accuracy when identifying images with occluded strawberries, but the real-time detection efficiency of the models was not mentioned, and experiments on model detection speed are still required to verify the feasibility of model integration into strawberry picking robots.

In summary, previous studies conducted valuable research on image segmentation of strawberry with different maturities to reduce the influence of environmental factors and improve the strawberry image segmentation accuracy, but have not considered improving the image segmentation accuracy without losing the model detection efficiency. Even though the image segmentation accuracy of the proposed method is acceptable, the recognition efficiency could not meet the real-time requirement of automatic strawberry picking. To address this issue, this paper proposes a method to improve DeepLabV3+ for strawberry image segmentation with different maturities, introducing the attention mechanism in the backbone feature extraction network and ASPP module, respectively,

and adjusting the weights of feature channels in the neural network propagation process through the attention mechanism, which could enhance the feature information of strawberry images with different maturities using fewer parameters, reducing the interference of environmental factors. The accuracy of strawberry image segmentation can be improved without losing the efficiency of model detection.

## 2. Materials and Methods

The overall workflow of this paper is shown in Figure 1, which mainly consists of 2 parts: (1) constructing the strawberry image dataset; (2) training the strawberry image segmentation model.

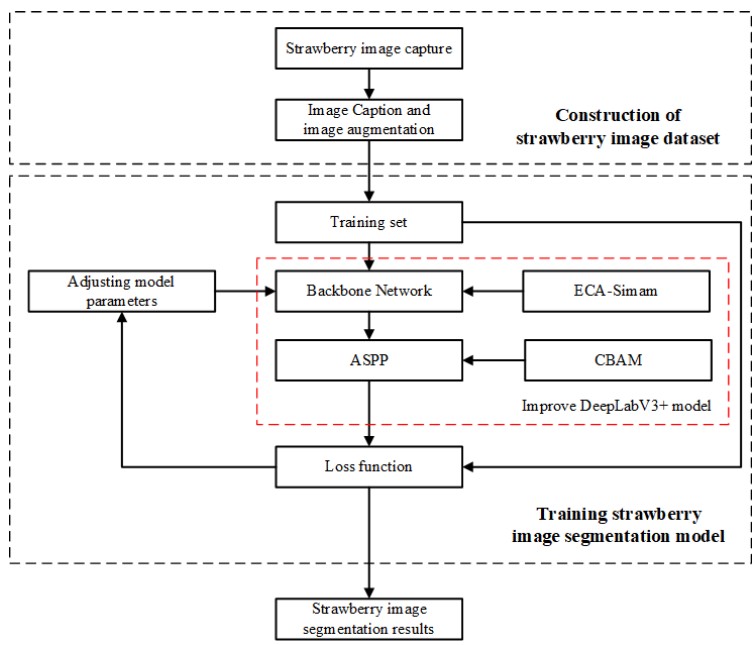

**Figure 1.** Overall workflow diagram.

### 2.1. Construction of Strawberry Image Dataset

Since there is no publicly available image dataset for strawberry cultivation on high shelves, this paper needed to construct an image dataset for strawberry cultivation on high shelves for research purposes. The strawberry image data in this paper were collected from the elevated strawberry cultivation base of Guoxin Modern Agriculture Company in Changchun, Jilin Province, China. A total of 1000 images of strawberries at different growth stages, different fruit numbers, different shading levels, and different light intensities were collected with a 40-megapixel HD camera, as shown in Figure 2, where Figure 2a–c show strawberry fruits at different growth stages and Figure 2d–f correspond to strawberry fruit images at different shading levels, respectively. During strawberry production, when the percentage of the red colored area is more than 75%, the fruit can be picked. Therefore, we classify the strawberry maturity into 3 stages: ripe (s > 75%), semi-ripe (25% < s < 75%), and unripe (s < 25%), according to the fruit coloring area (s), as shown in Figure 2a–c. The original image size of the acquired strawberry was $4624 \times 3468$ px, and to reduce the computational effort, we compressed the image size to $512 \times 512$ px during the training model.

The LabelMe [25] annotation tool was used to manually annotate each strawberry sample image to generate a Json file containing a large amount of label image information (location information of all annotated points, image size, and image category). We transformed each Json file into a masked label image by the Json command, as shown in Figure 3a. The immature strawberry mask image is green, the near-mature strawberry mask image is yellow, and the mature strawberry mask image is red, as shown in Figure 3b.

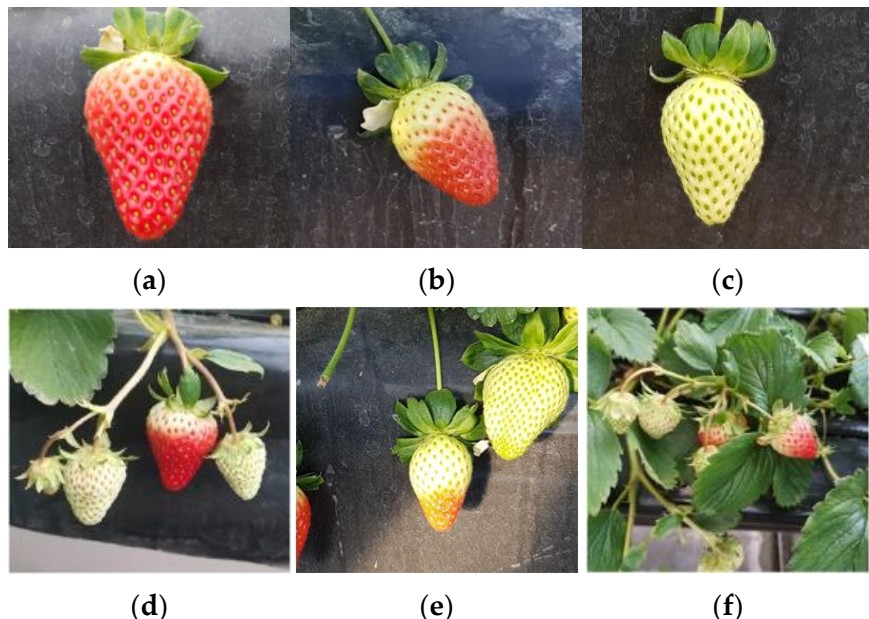

**Figure 2.** Images of strawberry samples collected under greenhouse environment. (**a**) Mature strawberry; (**b**) near-mature strawberry; (**c**) immature strawberry; (**d**) uncovered strawberry; (**e**) strawberry under strong light; (**f**) covered strawberry.

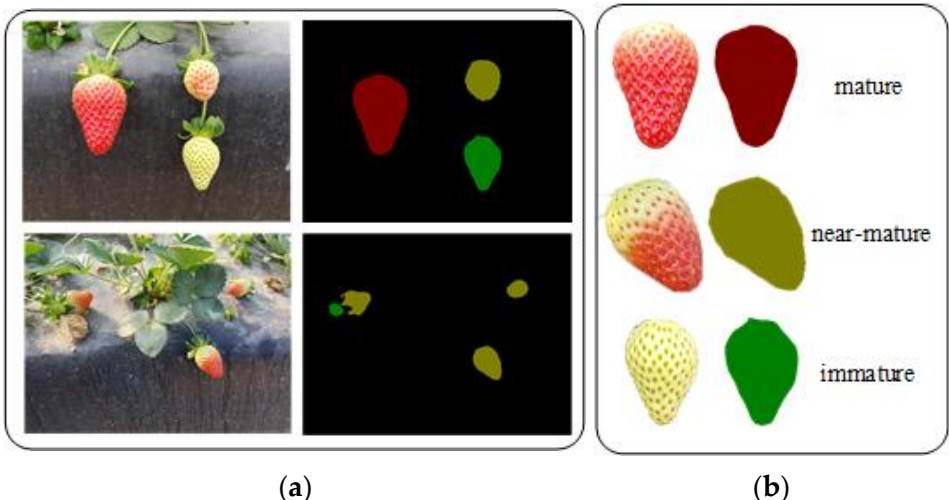

**Figure 3.** Labeled sample data set. (**a**) Strawberry image dataset and labeled images; (**b**) strawberry mask images.

To improve the network model training effect and model generalization ability, this study used the mirror flip method to increase the spatial diversity of strawberry images [26]; and brightness adjustment, adaptive contrast enhancement and Kmeans clustering were used to increase the diversity of strawberry image samples, as shown in Figure 4. A total of 6000 sample images were enhanced and the dataset was divided into training set (4200 images), test set (1200 images) and validation set (600 images) according to the ratio of 7:2:1.

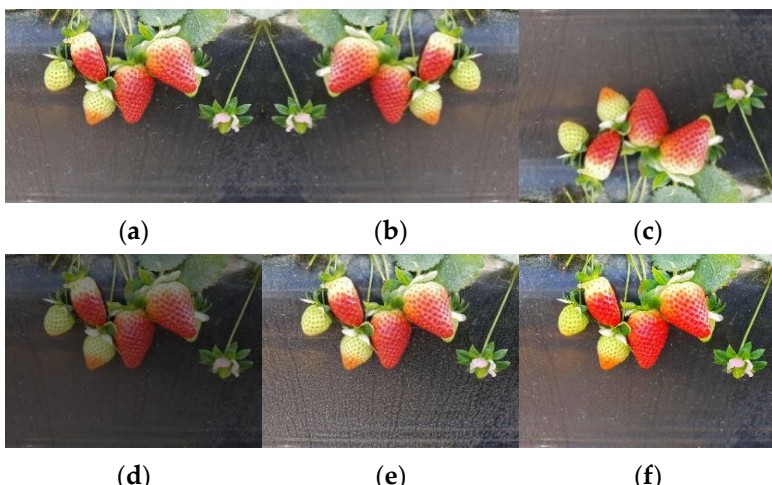

**Figure 4.** Plot of sample effects produced by different data enhancement methods. (**a**) Original image; (**b**) horizontal mirror; (**c**) vertical mirror; (**d**) diminished brightness; (**e**) contrast enhancement; (**f**) color enhancement.

### 2.2. Improved DeepLabV3+ Strawberry Image Segmentation Model

DeepLabV3+ [27] is a classical semantic segmentation network containing 2 parts: encoder and decoder. The encoder consists of a backbone feature extraction network and an atrous spatial pyramid pooling (ASPP) [28] structure, while the decoder obtains low-level features from the backbone feature extraction network and upsamples them to obtain pixel-by-pixel classification results of the same size as the input image. The backbone feature extraction network is an Xception [29] network, which is based on InceptionV3 [30] and uses depth wise separable convolution to replace the multi-size convolutional kernel feature response operation in InceptionV3, significantly reducing the number of model parameters, lowering the computational cost of the model, and improving the operational efficiency of the model. The ASPP structure consists of one $1 \times 1$ convolution in parallel, three $3 \times 3$ null convolutions and one global average pooling operation, where the expansion rates of the three $3 \times 3$ convolution operations are 6, 12, and 18, respectively. The structure is capable of multi-scale sampling of the feature map using null convolution operations with different sampling rates, expanding the perception of the convolution kernel, avoiding the loss of image detail features, and enhancing the adaptability to multi-scale targets. In the decoder part, the low-level features output from the backbone network are spliced and fused with the high-level features output from the encoder part, and then $1 \times 1$ convolution and upsampling are performed to obtain a classification result mask image with the same size as the input image.

The DeepLabV3+ network model was applied to strawberry image segmentation, and its model structure and segmentation effect are shown in Figure 5. Although the model can segment the strawberry image regions well, there are mis-segmentation cases when classifying strawberry images with different maturities, and the overall segmentation accuracy is low. Since the backbone network Xception and ASPP modules in DeepLabV3+ model adopt depthwise separable convolution and Dilated Convolutions with different expansion rates (6, 12, 18), respectively, although it improves the overall operational efficiency of the model, it reduces the extraction ability of target features and affects the association between local features of the target, thus producing the phenomenon of target semantic segmentation void; when DeepLabV3+ model deepens the number of network layers in the forward propagation process, the representation of obscured strawberry fruit features becomes weaker and weaker, leading to the disappearance of strawberry detail features in the whole network propagation process and the phenomenon of missegmentation.

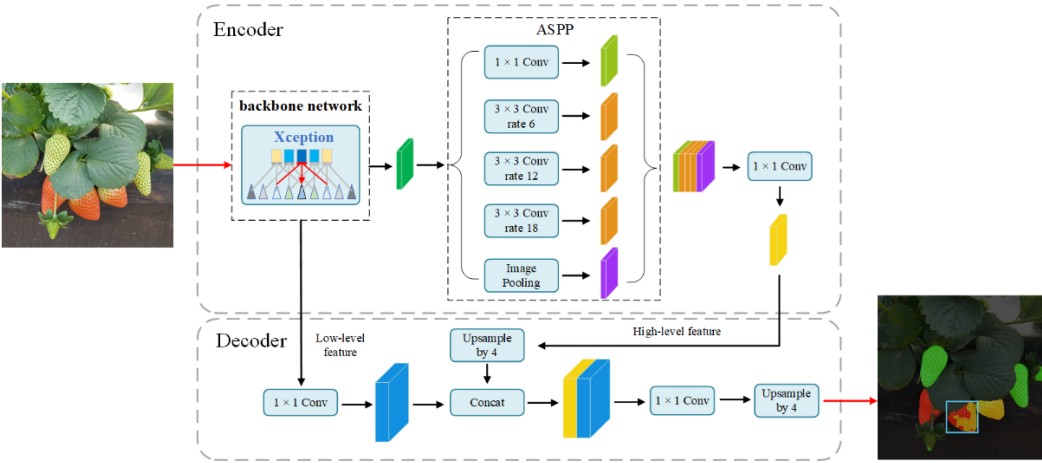

**Figure 5.** DeepLabV3+ structure and segmentation effect.

To address the above problems and considering the deployment of a strawberry image segmentation model in strawberry picking robots, the segmentation accuracy of the model needs to be improved without losing model operation efficiency. Therefore, this paper introduces the attention mechanism into the backbone network and ASPP module of the DeepLabV3+ model, respectively, and proposes an improved DeepLabV3+ strawberry image segmentation method. The structure of the improved DeepLabV3+ network is shown in Figure 6.

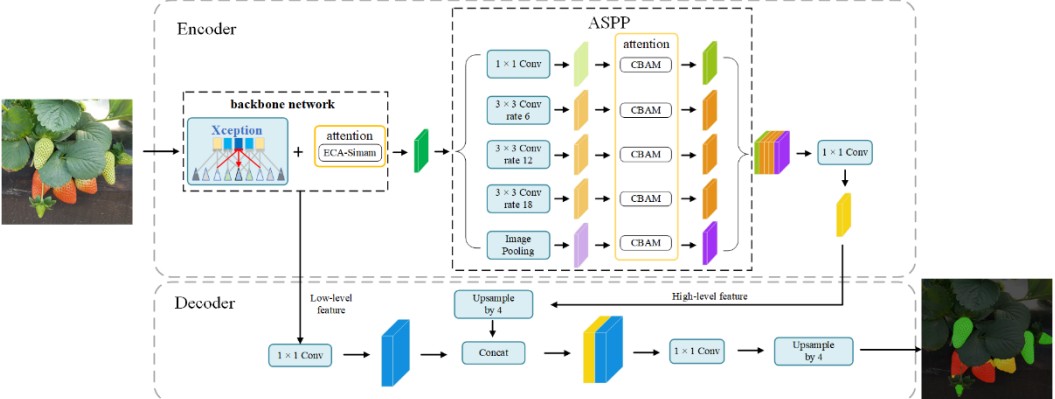

**Figure 6.** Improved DeepLabV3+ network structure diagram.

Compared with the classical DeepLabV3+ model, the present study mainly improves the backbone network and ASPP module in the DeepLabV3+ model. In the backbone network of the DeepLabV3+ model, after each depth-separable convolution, the ECA-SimAM module was introduced to strengthen the correlation between the feature information extracted by the network in the spatial domain and the channel domain, which improved the feature extraction ability of the backbone network and the accuracy of the model image segmentation. With the ASPP module of DeepLabV3+, after each atrous convolution, the CBAM module was introduced to adjust the weight share of feature channels to reduce the interference of environmental factors on strawberry image detection. The results showed an improvement in image segmentation accuracy.

### 2.2.1. Dual-Attention Mechanism to Optimize the Backbone Network

In this paper, the dual-attention mechanism is introduced into the DeepLabV3+ backbone network Xception to optimize the feature extraction ability of the backbone network, mine the important information in the feature map with fewer parameter calculations, and

adjust the proportion of important information weights in the neural network propagation process so as to enhance the feature extraction ability of the backbone network.

The network structure of the dual-attention mechanism is shown in Figure 7. The SimAM [31] module mines the importance of each neuron of the feature map by an energy function; no additional parameters are required to derive 3D attention weights for the feature map. Therefore, the ECA [32] module was combined with the SimAM module to construct an ECA-SimAM serial structure to improve the feature extraction capability of the ECA module without increasing the number of extra parameters for this model. The structure compresses the feature map F into a one-dimensional feature vector by global average pooling, then multiplies the convolved one-dimensional feature vector with the original input feature map F to get the feature map F′; then, it mines the importance of each neuron in the feature map F′ by a set of energy functions to derive a set of 3D attention weights for the feature map F′, and finally multiplies the obtained weights with the feature map F′ to get the final output feature map F″. The energy function equation is as follows:

$$e_t(w_t, b_t, y, x_i) = \frac{1}{M-1}\sum_{i=1}^{M-1}(-1-(\omega_t x_i + b_t))^2 + (1-(\omega_t t + b_t))^2 + \lambda\omega_t^2 \quad (1)$$

where $t$ and $x_i$ denote the target neurons and other neurons of the input features, $i$ denotes the index on the spatial dimension, $M$ denotes the number of neurons on the channel, and $\omega_t$ and $b_t$ denote the weights and biases of the neurons when they are transformed, respectively.

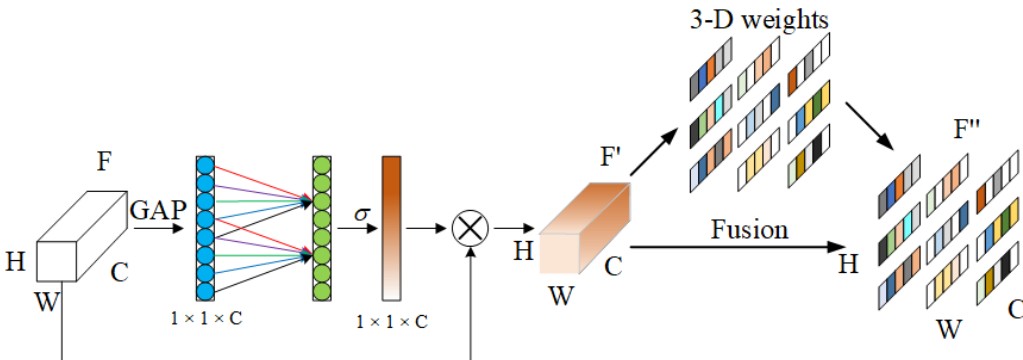

**Figure 7.** ECA-SimAM attention module.

The dual-attention mechanism optimizes the backbone network as shown in Figure 8, where the black font indicates the original structure of the backbone Xception network and the red font indicates the access attention module. Xception consists of Entry flow, Middle flow and Exit flow, each of which makes extensive use of deep separable convolution to split the correlation between the spatial dimension and the channel dimension, reducing the number of parameters needed for convolution calculations, reducing the complexity of parameter calculations, making the model more lightweight and losing some detection accuracy. Therefore, after each depth-separable calculation of the Xception network, the ECA-SimAM module is introduced to strengthen the correlation of the network extracted feature information in the spatial and channel domains, thus enhancing the feature extraction capability of the Xception network and providing model image segmentation accuracy.

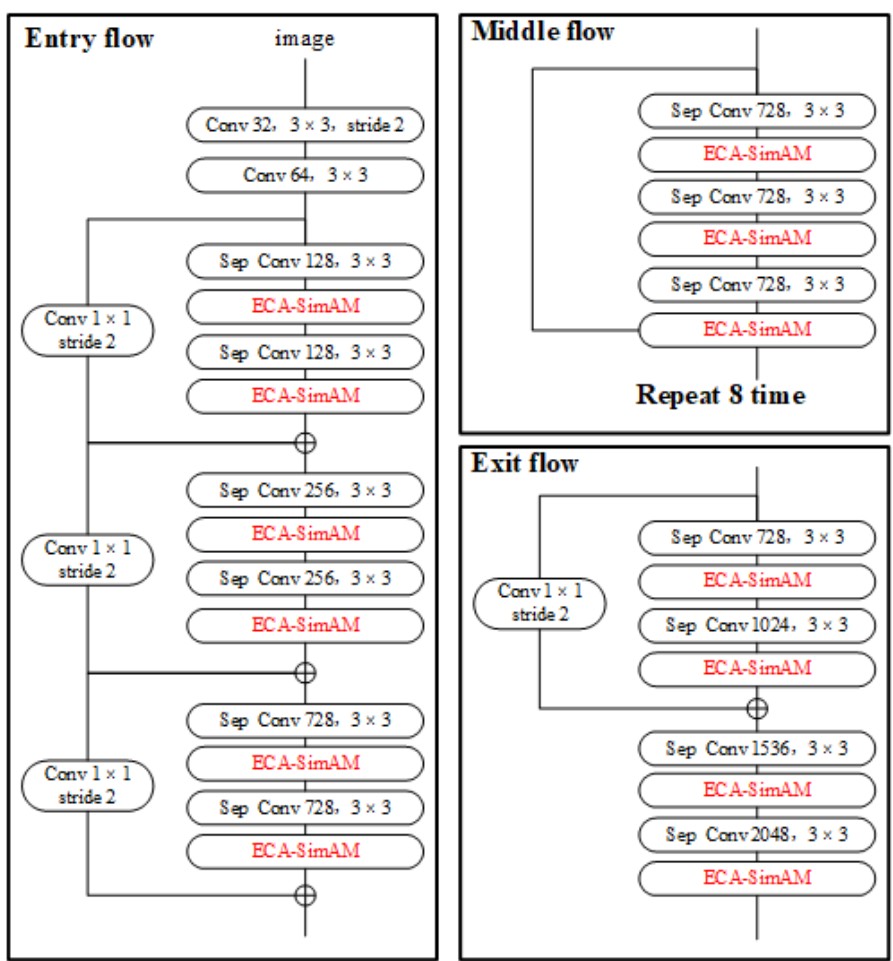

**Figure 8.** Dual-attention mechanism to optimize the backbone network.

### 2.2.2. Convolutional Attention Mechanism to Enhance ASPP

The ASPP module of the DeepLabV3+ model uses the null convolution with different expansion rates (6, 12, 18) to process the feature maps output from the backbone network in parallel to extracting multi-scale target information, but too large expansion rates will prevent the network from extracting image edge feature information well and also affect the association between local features of the target, making the expression of the obscured strawberry image weaker and reducing the strawberry image segmentation accuracy. Therefore, this paper integrates the CBAM [33] module into the ASPP module, reduces the interference of environmental factors (fruit adhesion, branch and leaf occlusion, and fruit stacking) through the attention mechanism, adjusts the weight share of the feature channels, solves the problem of weak feature expression of the occluded strawberry images, and improves the accuracy of model image segmentation. Among them, the CBAM module is an attention module combining both channel and spatial dimensions, and its network structure is shown in Figure 9; in the channel dimension, the input feature map F is pooled by the maximum and average values to get two sets of one-dimensional vectors; then, the multilayer perceptron network (MLP) is sequentially downscaled and upscaled from the channel dimension, the one-dimensional vector Mc(F) is obtained by summing up the elements, and finally Mc(F) is multiplied with the input The feature map $F'$ is obtained by multiplying Mc(F) with the input feature map F; in the spatial dimension, the feature map $F'$ is obtained by average pooling and maximum pooling to obtain 2 feature maps, then the feature map Mc($F'$) is obtained by descending the $3 \times 3$ convolution kernel, and finally the output feature map $F''$ is obtained by multiplying it with the feature map $F'$.

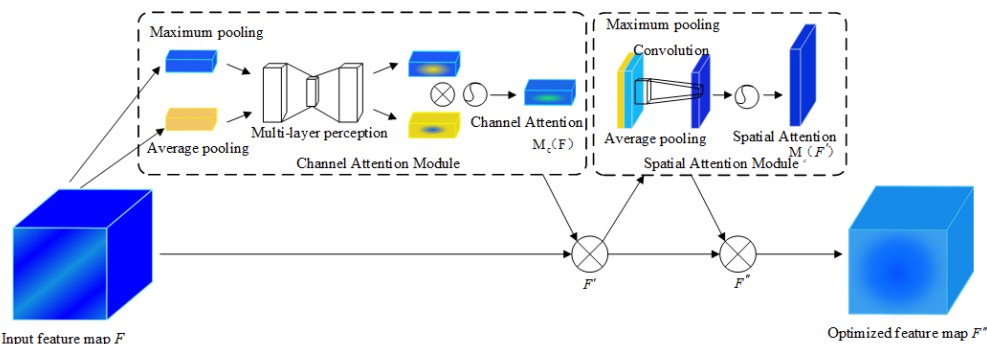

**Figure 9.** CBAM Attention Module.

### 2.3. Model Evaluation Metrics

In this study, strawberry ripeness was classified into mature (s > 75%), near-mature (25% < s < 75%) and immature (s < 25%) based on the strawberry fruits red coloring area (s). The Mean Intersection over Union ($e_{mIoU}$) and the Mean Pixel Accuracy ($e_{mPA}$) [34] were selected to evaluate the segmentation accuracy of the strawberry image segmentation model, and the Frames Per Second (FPS) was selected to evaluate the detection efficiency of the strawberry image segmentation model, where $e_{mIoU}$ can evaluate both missed and false detection, $e_{mPA}$ mainly evaluates the missed detection of the algorithm, and FPS indicates the number of input image frames per second. The specific formulas of $e_{mIoU}$ and $e_{mPA}$ are listed as:

$$e_{mIoU} = \frac{1}{k+1} \sum_{i=0}^{k} \frac{p_{ii}}{\sum_{j=0}^{k} p_{ij} + \sum_{j=0}^{k} p_{ji} - p_{ii}} \tag{2}$$

$$e_{mPA} = \frac{1}{k+1} \sum_{i=0}^{k} \frac{p_{ii}}{\sum_{i=0}^{k} p_{ij}} \tag{3}$$

where $k$ is the number of categories; $p_{ij}$ is the pixel marked as class $i$, but the prediction result is class $j$; $p_{ii}$ is the pixel marked as class $i$, and the prediction result is also class $i$.

## 3. Results

### 3.1. Confirmation of Test Environment and Parameter Settings

The test environment in this paper was divided into the hardware environment and the software environment, as shown in Table 1.

**Table 1.** Test environment configuration.

| Environment | Configuration | Parameters |
|---|---|---|
| Hardware environment | CPU | 4-core Intel(R) Xeon(R) Silver 4110 @ 2.10 GHz |
| | GPU | RTX 2080 Ti |
| | Memory | 16 GB |
| Software environment | System | Ubuntu 16.04 |
| | deep learning framework | Pytorch 1.10.0 |
| | programming environment | python 3.8 |
| | GPU parallel computing architecture | Cuba 10.1 |
| | GPU acceleration library | Cudnn 7.6.5. |

The model uses pre-trained weights from the PASCAL VOC dataset with a freeze training strategy. First, the backbone network parameters are frozen for training 1000 times, with 8 images in each batch, and the initial learning rate is set to 0.0003; then the network

training is unfrozen for 2000 times, with 2 images in each batch, and the initial learning rate is set to 0.00001, for a total of 3000 iterations.

### 3.2. Analysis of Strawberry Image Segmentation Models

#### 3.2.1. Ablation Experiments

In this paper, we mainly introduce the attention mechanism to improve the classical DeepLabV3+ network, optimize the backbone network, and enhance the ASPP module, respectively, and to improve the model segmentation accuracy without losing the model real-time detection efficiency. The model performance evaluation is shown in Table 2, where "$\sqrt{}$" indicates that the marked module is used for the experiment and no "$\sqrt{}$" indicates that this module is not added.

**Table 2.** Ablation experiments were performed by adding an attention module.

| DeeplabV3+ | SimAM Optimize Backbone Network | ECA Optimize Backbone Network | CBAM Enhanced ASPP | Network Model Parameters | MIoU | MPA | FPS |
|---|---|---|---|---|---|---|---|
| $\sqrt{}$ | | | | 209.70M | 74.04% | 85.45% | 8.52 |
| $\sqrt{}$ | $\sqrt{}$ | | | 209.71M | 75.12% | 86.40% | 8.06 |
| $\sqrt{}$ | | $\sqrt{}$ | | 209.74M | 76.33% | 86.33% | 8.18 |
| $\sqrt{}$ | | | $\sqrt{}$ | 209.77M | 75.92% | 86.59% | 8.37 |
| $\sqrt{}$ | $\sqrt{}$ | $\sqrt{}$ | | 209.75M | 80.78% | 88.55% | 7.71 |
| $\sqrt{}$ | $\sqrt{}$ | | $\sqrt{}$ | 209.77M | 77.52% | 87.44% | 7.83 |
| $\sqrt{}$ | | $\sqrt{}$ | $\sqrt{}$ | 209.80M | 78.63% | 87.74% | 8.01 |
| $\sqrt{}$ | $\sqrt{}$ | $\sqrt{}$ | $\sqrt{}$ | 209.81M | **83.05%** | **90.90%** | 7.67 |

Note: Bold is the optimal result and also the network proposed in this paper.

In the ablation experiment, compared with the classical DeepLabV3+, DeepLabV3+ with the SimAM module, DeepLabV3+ with the ECA module and DeepLabV3+ with the ECA-SimAM module, the MIoU was improved by 1.08%, 2.29% and 6.74%, respectively, and the MPA was improved by 0.95%, 0.88%, and 3.1%, respectively. The experimental results show that the DeepLabV3+ model with the ECA-SimAM module added has the highest MIoU and MPA, and the dual-attention mechanism has stronger feature extraction capability than the single-attention mechanism.

To further analyze the improvement of the model image segmentation accuracy, different attention mechanisms were introduced for experiments, respectively. The experimental results show that the MIoU and MPA of DeepLabV3+ with the introduction of the ECA-SimAM module and CBAM module are the highest, which are 83.05% and 90.90%, respectively, and compared with the classic DeepLabV3+ model; MIou and MPA are improved by 9.01% and 5.62%, respectively, compared with the classical DeepLabV3+ model. Meanwhile, the FPS of the model running in real time is 7.67, which is only 0.85 lower compared with the classical DeepLabV3+ model, and does not affect the efficiency of the model running in real time. Overall, the model proposed in this paper can improve segmentation accuracy without losing real-time detection efficiency.

#### 3.2.2. Performance Analysis of Different Models

To verify the superiority of the model in this paper for the segmentation accuracy of strawberry images with different maturity, U-Net [35], PSPNet [36], HRNet [37] and DeepLabV3+ models were selected for comparison tests, and the evaluation indexes of the segmentation accuracy of different models were obtained as shown in Table 3.

**Table 3.** Segmentation accuracy of different segmentation models in strawberry images.

| Models | Intersection over Union | | | $e_{mIoU}$ | Pixel Accuracy | | | $e_{mPA}$ |
|--------|----------|-------------|--------|------------|----------|-------------|--------|------------|
| | Immature | Near-Mature | Mature | | Immature | Near-Mature | Mature | |
| U-Net | 74.14% | 70.87% | 74.65% | 73.22% | 83.71% | 84.92% | 78.83% | 82.48% |
| PSPNet | 72.66% | 72.84% | 75.02% | 73.50% | 82.60% | 89.18% | 79.59% | 83.79% |
| HRNet | 77.99% | 71.32% | 77.21% | 75.51% | 84.22% | 83.92% | 87.97% | 85.37% |
| DeepLabV3+ | 76.95% | 69.91% | 75.26% | 74.04% | 86.54% | 88.21% | 81.59% | 85.45% |
| Proposed | 80.59% | 82.23% | 86.34% | **83.05%** | 90.40% | 90.52% | 91.79% | **90.90%** |

Note: Bold is the optimal result and also the network proposed in this paper.

As shown in the Table 3, the $e_{mIoU}$ and the $e_{mPA}$ in this paper are the highest, which are 83.05% and 90.90%, respectively. Compared with the U-Net, PSPNet, HRNet and DeepLabV3+ models, the $e_{mIoU}$ of the model in this paper is improved by 9.83%, 9.55%, 7.51% and 9.01%, respectively; the $e_{mPA}$ is improved by 8.42%, 7.11%, 5.53 and 5.45%, respectively. Overall, the performance of the model proposed in this paper is optimal.

3.2.3. Different Model Segmentation Results

The visualization results of the model in this paper with different image segmentation models are shown in Figure 10. In the visualization results, the red area is the ripe strawberry mask image, the yellow area is the semi-ripe strawberry mask image, and the green area is the unripe strawberry mask image; the blue boxed area in the figure indicates the mis-segmentation phenomenon of the model.

From Figure 10a, it can be seen that all models can accurately segment the strawberry images when the fruits are in the unobstructed situation, and only U-Net is inaccurate in segmentation of the details. From Figure 10b, it can be seen that U-Net segmentation is poor when there are adhesions between fruits, and it is difficult to accurately segment the strawberry images with different ripeness; PSPNet, HRNet, DeepLabV3+ and the models in this paper can accurately segment the strawberry images with different ripeness.

From Figure 10c, it can be seen that U-Net, PSPNet, HRNet, and DeepLabV3+ all showed missegmentation when the fruit was obscured by leaves. In sample 4, U-Net could not accurately segment the strawberry images, and PSPNet, HRNet and DeepLabV3+ had difficulty in accurately extracting the detailed features of strawberries of different maturity and failed to accurately segment the strawberry images of different maturity; in sample 5, U-Net and PSPNet incorrectly classified the ripeness of strawberries, and HRNet and DeepLabV3+ accurately classified the ripeness of strawberries, but failed to completely segment the strawberry images in the unobscured area, while the model in this paper can not only accurately classify the ripeness of strawberries but also completely segment the strawberry images.

From Figure 10d, we can see that U-Net, PSPNet, HRNet, and DeepLabV3+ all show mis-segmentation when the fruits are stacked, while the model in this paper can accurately segment the strawberry images.

In summary, compared with the classical image segmentation model, the model in this paper can effectively reduce the interference of environmental factors, extract more strawberry image feature information, accurately segment strawberry images with different maturity, and has stronger robustness.

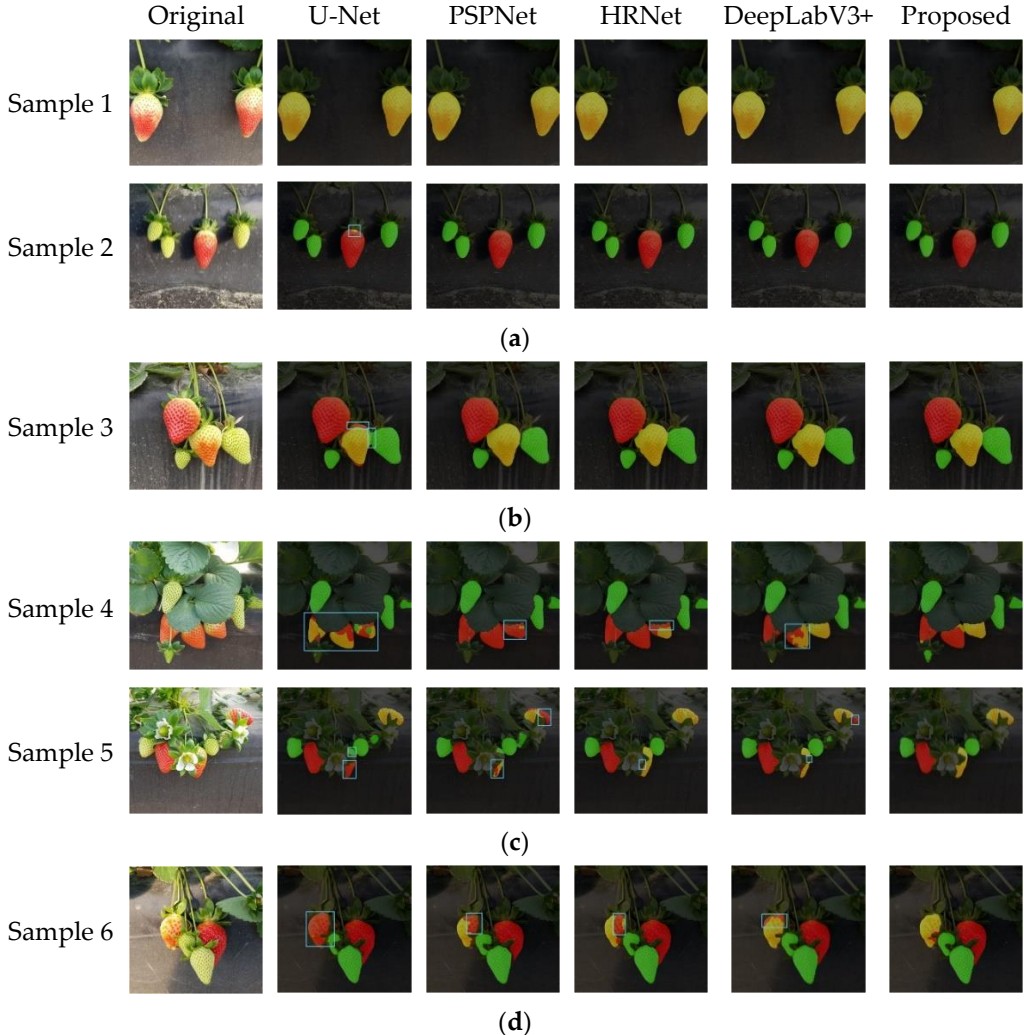

**Figure 10.** The effect of different strawberry image segmentation models. (**a**) No shade; (**b**) fruit adhesion; (**c**) leaf shading; (**d**) fruit stacking.

## 4. Discussion

Emerging in smart agriculture research, image segmentation of strawberries at different maturity levels has attracted much interest, which has focused on designing many image segmentation methods to reduce the interference of environmental factors and improve the segmentation accuracy of images. The ultimate goal of these studies was also to deploy the methods in strawberry picking robots to provide a basis for their accurate operation. Therefore, the research on image segmentation methods for strawberries should improve the image segmentation accuracy without losing the real-time operation efficiency of the model. In this paper, we propose an improved DeepLabV3+ method for strawberry image segmentation with different maturity levels, which is based on a DeepLabV3+ network and allows the neural network to ignore irrelevant feature information and focus on important information through the attention mechanism so as to improve the model image segmentation accuracy with less computational effort. The method can improve strawberry image segmentation accuracy on the basis of satisfying real-time operation of strawberry picking robots.

In the future work, we will conduct experiments on different locations and different varieties of strawberries, analyze the commonality and characteristics of different locations and different varieties of strawberries, and build a more universal strawberry image dataset to make the designed image segmentation model more accurate. At the same time, considering the deployment of the model in strawberry picking robots, the model should

be further designed to be lightweight, with reduced internal architecture and components and reduced computational cost, without losing its recognition accuracy, so as to be better adapted to strawberry picking robots.

## 5. Conclusions

In this paper, we propose a method to improve DeepLabV3+ for strawberry image segmentation with different maturity levels. The multi-attention mechanism is incorporated into the DeepLabV3+ model to increase the weight of strawberry image feature information and decrease the weight of environmental background feature information so that the network model pays more attention to the feature information with larger weight and ignores the feature information with smaller weight during the training process, effectively reducing the interference of environmental factors (dense distribution of fruits, leaf occlusion and fruit stacking) and improving the accuracy of strawberry image segmentation. The experimental results show that the proposed method can accurately segment strawberry images with different maturity, the average pixel accuracy and average intersection ratio of the model are 90.9% and 83.05%, respectively, and the FPS is 7.67.

The method is based on deep learning to extract digital information of strawberries, which accurately segmented images of strawberries with different ripeness. This method could be applied to other fruits and vegetables (tomatoes, cucumbers, etc.) after re-collecting their sample data, building the training data set and training the model to complete the image segmentation and ripeness judgment. This work provides a real case study of utilizing machine learning in digital agriculture practice.

**Author Contributions:** C.C., J.T. and P.Z. conceived and designed the manuscript. C.C., J.T., P.Z., Y.Y. and J.Z. analyzed the data and wrote the paper. Y.Y. and J.Z. reviewed the manuscript. All authors have read and agreed to the published version of the manuscript.

**Funding:** This research was funded by the Department of Science and Technology of Jilin Province, (20200402116NC).

**Institutional Review Board Statement:** Not applicable.

**Informed Consent Statement:** Not applicable.

**Data Availability Statement:** All data, models, and code generated or used during the study appear in the submitted article.

**Acknowledgments:** We are grateful to Changchun Guoxin Modern Agriculture Com-pany for providing the experimental site.

**Conflicts of Interest:** The authors declare no conflict of interest.

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
