# Peer review of "Determining Strawberries’ Varying Maturity Levels by Utilizing Image Segmentation Methods of Improved DeepLabV3+"

_agronomy, doi:10.3390/agronomy12081875_

Round 1

Reviewer 1 Report

The manuscript proposed a new method for improving strawberry image segmentation for maturity levels detection.

The article is really well presented, with the complete dataset of strawberries based on a real-world dataset.

However:

1. In the manuscript, the proposed "improve DeepLabV3" does not show any pseudocode, on which part of the DeepLabV3+ that have been improved?

2. Even on a quick reading, it is easy to spot inconsistent writing uniformity.

3. Also, the number behind the digit needs to be persistent.

Author Response

Comment 1: The manuscript proposed a new method for improving strawberry image segmentation for maturity levels detection.

The article is really well presented, with the complete dataset of strawberries based on a real-world dataset.

In the manuscript, the proposed "improve DeepLabV3" does not show any pseudocode, on which part of the DeepLabV3+ that have been improved?

Response: Thank you for careful reviewing our manuscript, we are grateful to your question. We have added a paragraph at the end of section 2.2, explained in detail of the main improvements of DeepLabV3+model with the backbone network and ASPP modules.

Comment 2:Even on a quick reading, it is easy to spot inconsistent writing uniformity.

Response: Thank you for your comments. Based on your comment, we noticed that the images in the text were not aligned with the left side of the text, which is not consistent in formatting. Therefore, We have repositioned all images so that they are aligned with the left side of the text and any inconsistence of writing uniformity.

Comment 3:Also, the number behind the digit needs to be persistent.

Response: Thank you for your comments. Based on your suggestion, We found that some of the letters after the image labels in the text should not be added in parentheses, which is not in accordance with the format of this publication. We have removed all brackets from the letters after the image labels.

Reviewer 2 Report

1. The paper is well written and results are presented in expected manner. Well done.

2. Author may check line 252 where some typo seems to be present. The written text - "Cudnn 7.6.5. 2The hardware", which does not make sense. Please correct.

3. In the introduction section, the strawberry fruit ripeness estimation using various approach like - hyperspectral, thermal, multispectral, RGB etc need to be discussed. This will also help authors to do comparison in the discussion section. Authors should spend at least one paragraph to add those studies. 

4. Strawberry fruit ripeness is also related to its water content. These scenarios can also be discussed in the introduction section in brief. 

Author Response

Comment 1:The paper is well written and results are presented in expected manner. Well done.

Author may check line 252 where some typo seems to be present. The written text - "Cudnn 7.6.5. 2The hardware", which does not make sense. Please correct.

Response: Thank you for pointing out typo issue. They occurred in subsection 3.1, where an extra value of "2" was entered in the text. It has been removed and corrected in the revised version.

Comment 2:In the introduction section, the strawberry fruit ripeness estimation using various approach like - hyperspectral, thermal, multispectral, RGB etc need to be discussed. This will also help authors to do comparison in the discussion section. Authors should spend at least one paragraph to add those studies.

Response: Thank you for your constructive comments. Based on your suggestion, We found that the study on spectroscopic analysis of strawberry ripening was not cited in the introduction. Therefore, We added the citation of the study on spectroscopic analysis of strawberry ripening in the second paragraph of the introduction.

Comment 3:Strawberry fruit ripeness is also related to its water content. These scenarios can also be discussed in the introduction section in brief.

Response: Thank you for your constructive comments. According to your suggestion, the correlation between water content and fruit ripeness of strawberries should be discussed in the introduction section. We cited literature [4] where not only fruit ripeness is discussed in relation to water content, but also strawberry ripeness was analyzed by spectroscopic techniques.

Reviewer 3 Report

Please elaborate the term FPS as it is first time introduced in the Abstract section. Revise the fourth sentence and the last sentence in the Introduction section for better understanding. What is the computation complexity of the dual attention mechanism on the improved DeepLabV3+ structure contemplating the operation efficiency? Reviewer suggests the section 3.2 should be placed before the Results section. The computer hardware environment and programming scenario presented in section 3.1 can be listed in a Table format.

Author Response

Comment 1:Please elaborate the term FPS as it is first time introduced in the Abstract section.

Response: Thank you for pointing this out. It is true that FPS is not described in detail, we have explained FPS in the revision, which stands for frames per second.

Comment 2:Revise the fourth sentence and the last sentence in the Introduction section for better understanding.

Response: Thank you for your constructive comments. According to the issue you raised, the fourth and last sentence in the introduction part of this paper is vague in description and not easy to understand. Therefore, we modified these parts in the revision.

Comment 3:What is the computation complexity of the dual attention mechanism on the improved DeepLabV3+ structure contemplating the operation efficiency?

Response: Thank you for your question. In subsection 2.2.1 of this paper, the dual attention mechanism in the improved DeepLabV3+ architecture consists mainly of a serial structure consisting of an ECA module and a SimAM module. Among them, the SimAM module differs from the existing channel/space attention module in that it can mine the importance of each neuron through the energy function without additional parameters to derive the 3D attention weights of the feature map. we combined the ECA module with the SimAM module to improve the feature extraction capability of the ECA module without increasing the number of extra parameters of the model.

Comment 4:Reviewer suggests the section 3.2 should be placed before the Results section.

Response: Thank you very much for your constructive comments and kind suggestion. We moved section 3.2 before the results and it becomes section 2.3.

Comment 5:The computer hardware environment and programming scenario presented in section 3.1 can be listed in a Table format.

Response: Thank you very much for your constructive comments and suggestion. Describing the computer hardware environment and programming environment have been modified table form.